# Prevalence and Clinical Associations of Malnutrition and Sarcopenia Risk in Gastroenterology Inpatients: A Multicenter Cross-Sectional Study in Turkey

**DOI:** 10.3390/diagnostics15222935

**Published:** 2025-11-20

**Authors:** Göksel Bengi, Süleyman Dolu, Yavuz Özden, Nevin Oruç, Mukaddes Tozlu, Gözde Derviş Hakim, Genco Gençdal, Ali Rıza Çalışkan, Müge Ustaoğlu, Ufuk Kutluana, Engin Altıntaş, Galip Egemen Atar, Ahmet Uyanıkoğlu, Sezgin Barutçu, Kader Irak, Deniz Koç, Berat Ebik, Züleyha Akkan Çetinkaya, Haluk Tarık Kani, Dilek Oğuz, Filiz Araz, Altay Kandemir, Nermin Mutlu Bilgiç, Özdal Ersoy, Özlem Gül, Banu Kara, Burak Özşeker, Hüseyin Alkım, Sedat Boyacıoğlu, Ayşe Kefeli, Hasan Yılmaz, Pembe Keskinoğlu, Yasemin Gökden Gök, Çağlayan Keklikkıran, Müjde Soytürk, Salih Tokmak, Murat Aladağ, Hakan Ünal, Funda Uğur Kantar, Yusuf Serdar Sakin, Meral Kayhan, Ozan Cengiz, Tolga Gözmen, İsmail Atasoy, Hale Akpınar

**Affiliations:** 1Department of Gastroenterology, Dokuz Eylul University Faculty of Medicine, 35330 Izmir, Türkiye; soyturkmuj@gmail.com (M.S.); ismail.atasoy@deu.edu.tr (İ.A.); haleakpinar05@gmail.com (H.A.); 2Department of Gastroenterology, Kayseri City Hospital, 38080 Kayseri, Türkiye; yavuzozden@gmail.com; 3Department of Gastroenterology, Ege University Faculty of Medicine, 35040 Izmir, Türkiye; nevinoruc@hotmail.com; 4Department of Gastroenterology, Sakarya Training and Research Hospital, 54100 Sakarya, Türkiye; mukaddes_tozlu@hotmail.com; 5Department of Gastroenterology, Izmir Faculty of Medicine, University of Health Sciences, Izmir City Hospital, 35540 Izmir, Türkiye; gozdedervis@gmail.com (G.D.H.); tolga.gozmen@deu.edu.tr (T.G.); 6Department of Gastroenterology, Koç University Faculty of Medicine, 34450 Ankara, Türkiye; ggencdal@ku.edu.tr; 7Department of Gastroenterology, Adiyaman Training and Research Hospital, 02040 Adiyaman, Türkiye; acaliskan@adiyaman.edu.tr; 8Department of Gastroenterology, Ondokuz Mayis University Faculty of Medicine Hospital, 55270 Samsun, Türkiye; muge.ustaoglu@omu.edu.tr; 9Department of Gastroenterology, Pamukkale University Faculty of Medicine Hospital, 20070 Denizli, Türkiye; ukutluana@pau.edu.tr; 10Department of Gastroenterology, Mersin University Faculty of Medicine Hospital, 33110 Mersin, Türkiye; enginaltintas@gmail.com; 11Department of Gastroenterology, Antalya Training and Research Hospital, 07100 Antalya, Türkiye; glpegemen@hotmail.com; 12Department of Gastroenterology, Sanliurfa University Faculty of Medicine, 63200 Sanliurfa, Türkiye; auyanikoglu@hotmail.com; 13Department of Gastroenterology, SANKO University Hospital, 27090 Gaziantep, Türkiye; sezginbarutcu@hotmail.com; 14Department of Gastroenterology, Basaksehir Cam and Sakura City Hospital, 34480 Istanbul, Türkiye; dr.kaderirak@hotmail.com; 15Department of Gastroenterology, Istanbul Gaziosmanpasa Taksim Training and Research Hospital, 34255 Istanbul, Türkiye; drdenizkoc@gmail.com; 16Department of Gastroenterology, Diyarbakir Gazi Yasargil Training and Research Hospital, 21070 Diyarbakir, Türkiye; beratebik@gmail.com; 17Department of Gastroenterology, Istanbul Acibadem Atasehir Hospital, 34758 Istanbul, Türkiye; zuleyha.akkan@acibadem.com; 18Department of Gastroenterology, Marmara University School of Medicine, 34899 Istanbul, Türkiye; tarikkani@marmara.edu.tr; 19Department of Gastroenterology, Kirikkale University Faculty of Medicine, 71450 Kirikkale, Türkiye; ddkoguz@yahoo.com; 20Department of Gastroenterology, Baskent University Adana Dr. Turgut Noyan Application and Research Center, 01100 Adana, Türkiye; filizaraz@baskent.edu.tr; 21Department of Gastroenterology, Aydin Adnan Menderes University Faculty of Medicine, 09100 Aydin, Türkiye; altaykandemir@yahoo.com; 22Department of Gastroenterology, Umraniye Training and Research Hospital, 34764 Istanbul, Türkiye; drnerminmutlu@yahoo.com.tr; 23Department of Gastroenterology, Acibadem University Atakent Hospital, 34303 Istanbul, Türkiye; ozdalersoy@gmail.com; 24Department of Gastroenterology, Lokman Hekim University Ankara Hospital, 06510 Ankara, Türkiye; ozlem.gul@lokmanhekim.edu.tr; 25Department of Gastroenterology, University of Health Sciences Adana City Training and Research Hospital, 01370 Adana, Türkiye; banu.kara@sbu.edu.tr; 26Department of Gastroenterology, Mugla University Faculty of Medicine, 48000 Mugla, Türkiye; burakozseker@mu.edu.tr; 27Department of Gastroenterology, University of Health Sciences, Sisli Hamidiye Etfal Training and Research Hospital, 34371 Istanbul, Türkiye; huseyin.alkim@sbu.edu.tr; 28Department of Gastroenterology, Baskent University Faculty of Medicine, 06490 Ankara, Türkiye; sboyacioglu@baskent.edu.tr; 29Department of Gastroenterology, Tokat Gaziosmanpasa University Faculty of Medicine, 60250 Tokat, Türkiye; ayse.kefeli@gop.edu.tr; 30Department of Gastroenterology, Kocaeli University Faculty of Medicine, 41380 Kocaeli, Türkiye; hasan.yilmaz@kocaeli.edu.tr; 31Department of Biostatistics and Medical Informatics, Dokuz Eylul University Faculty of Medicine, 35340 Izmir, Türkiye; pembe.keskinoglu@deu.edu.tr; 32Department of Internal Medicine, Istanbul Prof. Dr. Cemil Tascioglu City Health Application and Research Center, 34384 Istanbul, Türkiye; yasemin.gokden@sbu.edu.tr; 33Department of Gastroenterology, Recep Tayyip Erdogan University Faculty of Medicine, 53200 Rize, Türkiye; caglayan.keklikkiran@erdogan.edu.tr; 34Department of Gastroenterology, Istanbul Medical Park Hospitals Group, 34180 Istanbul, Türkiye; salihtokmak@duzce.edu.tr; 35Department of Gastroenterology, Bezmialem Vakif University Medicine Faculty, 34093 Istanbul, Türkiye; murat.aladag@inonu.edu.tr; 36Department of Gastroenterology, Acibadem University Faculty of Medicine, 34752 Istanbul, Türkiye; umit.unal@acibadem.com; 37Department of Gastroenterology, Bakircay University Cigli Training and Research Hospital, 35620 Izmir, Türkiye; funda.ugurkantar@bakircay.edu.tr; 38Department of Gastroenterology, Private Acibadem Izmir Kent Hospital, 35630 Izmir, Türkiye; yusuf.sakin@acibadem.com; 39Department of Internal Medicine, University of Health Sciences, Gastroenterology Clinic, Ankara City Hospital, 06800 Ankara, Türkiye; meralakdogan.kayhan@sbu.edu.tr; 40Department of Gastroenterology, Usak University Faculty of Medicine, 64060 Usak, Türkiye; drozancengiz@gmail.com

**Keywords:** malnutrition, sarcopenia, nutritional risk

## Abstract

**Background/Objectives**: This study aimed to determine the prevalence of malnutrition and sarcopenia risk among patients hospitalized in gastroenterology clinics across different geographical regions of Turkey, to identify their risk factors, and to evaluate their associations with clinical outcomes. **Methods**: A total of 1051 patients admitted to 36 gastroenterology clinics across six geographical regions of Turkey during the week of 14 November 2024 were evaluated in a cross-sectional design. The nutritional status of the patients was assessed using the NRS-2002 questionnaire, while the risk of sarcopenia was evaluated with the SARC-F questionnaire. Demographic data, clinical diagnoses, disease severity scores, and comorbidities were also recorded and analyzed. **Results**: Of the patients included in the study, 54.7% were female, and the mean age was 61.7 ± 17.2 years. The prevalence of malnutrition risk was 27.8%, while the prevalence of sarcopenia risk was 32.7%. Patients with malnutrition risk had a lower BMI (24.7 ± 5.3 vs. 27.1 ± 5.4, *p* < 0.001) and were older (67.6 ± 16.0 vs. 56.5 ± 17.1, *p* < 0.001). The risks of sarcopenia and malnutrition were significantly higher in patients with liver cirrhosis (40.7% malnutrition; 54.5% sarcopenia), gastrointestinal malignancy (50.5%; 44.2%), and diabetes mellitus. Logistic regression analysis identified older age, male sex, and presence of malignancy as independent risk factors for malnutrition, whereas older age, female sex, presence of malnutrition, liver cirrhosis, and heart failure were independent risk factors for sarcopenia. A strong correlation was also found between malnutrition and sarcopenia (r = 0.544, *p* < 0.001). **Conclusions**: Approximately one-third of patients hospitalized in gastroenterology clinics across Turkey are at risk of malnutrition and sarcopenia. These conditions are particularly associated with malignancy, cirrhosis, and metabolic comorbidities. Our findings highlight the necessity of systematic nutritional and sarcopenia screening upon hospital admission.

## 1. Introduction

Recognition of malnutrition and provision of adequate nutritional support in hospitalized patients constitute an essential component of treating the underlying disease. The prevalence of malnutrition among newly admitted hospital patients varies depending on the center, patient population, and assessment methods used, but it is generally reported to be between 30% and 50% [1,2,3]. Changes occurring in the gastrointestinal system (GIS) of malnourished patients, along with impaired immune function, lead to disturbances in digestion and absorption [4,5]. Although malnutrition tends to become more pronounced during hospitalization, it is also an independent predictor of nosocomial infections and complications. Furthermore, malnutrition is closely associated with prolonged hospital stays and increased mortality [6,7]. Providing appropriate nutritional support to hospitalized patients has been shown to reduce mortality and complication rates and to shorten the duration of hospitalization [8].

The causes of malnutrition are usually multifactorial and may result from both the metabolic effects of the underlying disease and reduced food intake. In addition, factors such as age, therapeutic interventions, education level, and/or low socioeconomic status may further increase the risk of malnutrition [9]. Despite its high prevalence, malnutrition is rarely recognized by clinicians. The main challenges in identifying patients at risk include lack of awareness and the time required for screening and implementation of nutritional therapy [10]. Nutritional risk screening is an important first step in the nutritional care process; however, it is unfortunately not routinely performed in healthcare centers in Turkey. Recently, a multicenter retrospective study including 5051 patients from 12 countries across Europe and the Middle East identified nutritional risk in 32.6% of patients, with considerable heterogeneity across centers, regions, and departments [11]. In Turkey, a multicenter study conducted in 2009 using the Nutritional Risk Screening-2002 (NRS-2002) assessment form reported a 15% prevalence of malnutrition risk among hospitalized patients at the time of admission. In that study, being over 60 years of age was identified as the only significant nutritional risk factor, while other parameters were not examined [12]. Similar previous studies conducted in Turkey were mostly single-center with limited sample sizes, and nutritional status in hospitalized patients was assessed using different methods, resulting in widely varying reported malnutrition rates.

Sarcopenia refers to the loss of muscle mass and function. Although it is often a part of the aging process, it can also occur as secondary sarcopenia due to pathogenic mechanisms related to disease activity or protein deficiency associated with nutrition [13]. Sarcopenia may remain asymptomatic until it becomes severe and is often undetected. Changes in body composition can be difficult to identify due to obesity, alterations in fat mass, or edema. The prevalence of sarcopenia in hospitalized patients has been reported to range widely from 10% to 69.5%. Sarcopenia increases the risk of adverse outcomes such as physical disability, reduced quality of life, and mortality [14,15].

The aim of the present study, unlike previous ones, is to determine the prevalence of malnutrition and the risk of sarcopenia among patients hospitalized in gastroenterology clinics of hospitals located in various cities of Turkey with different geographical and sociocultural characteristics, and to identify the factors that may contribute to the increase in these risks.

## 2. Materials and Methods

### 2.1. Patients

All patients hospitalized with various diagnoses in the gastroenterology clinics of 36 hospitals across 22 cities representing different geographical regions of Turkey during the week of 14 November 2024, were included in the study. Since these hospitals are healthcare institutions where approximately 90% of gastroenterology patients are hospitalized and followed up, the study data are representative of gastroenterology inpatients at the national level.

A sample size calculation was performed for the primary outcome variables—malnutrition and sarcopenia rates. For malnutrition, based on the study by Korfalı et al. [12], in which the frequency of malnutrition among gastroenterology inpatients was reported as *p* = 19.1%, the required sample size was calculated as *n* = 766 with a precision (d) of 0.05, 95% power, and a 95% confidence interval. For sarcopenia, using the data from the study by Özer et al. [16], which reported the prevalence of sarcopenia in hospitalized patients as *p* = 48%, the required sample size was calculated as *n* = 1018 under the same parameters (d = 0.05, 95% power, 95% confidence interval). A total of 1051 patients were included in the present study, thereby meeting the sample size requirements for both outcome variables.

Patients aged 18 years and older who underwent malnutrition and sarcopenia risk screening within the first 48 h of hospitalization were included in the study. Patients with a hospital stay of only one day, those with a history of hospitalization within the past three months, pregnant women, and patients admitted to the intensive care unit were excluded from the study.

This study protocol was prepared by the Nutrition Working Group of the Turkish Society of Gastroenterology (TGD) in October 2024, and the data collection forms were distributed to all members via the official TGD email address. All participating centers collected data from hospitalized patients during a predetermined one-week period (including 14 November, World Nutrition Day) using the previously prepared forms. Patient data were obtained from hospital automation systems and medical records. Using a standardized data collection form, demographic information (age, sex, body mass index), admission diagnoses, disease duration, and disease severity were recorded. To assess disease severity, the following scoring systems were used: the Bedside Index of Severity in Acute Pancreatitis (BISAP) for acute pancreatitis, the Harvey–Bradshaw Index (HBI) for Crohn’s disease, the Mayo score for ulcerative colitis, the Child classification for liver cirrhosis, the Glasgow–Blatchford score for upper gastrointestinal bleeding, the Oakland score for lower gastrointestinal bleeding, and the Eastern Cooperative Oncology Group (ECOG) performance score for gastrointestinal cancers.

### 2.2. Malnutrition Risk Assessment

The nutritional status of the patients was evaluated using the NRS-2002 screening test. This test was chosen because its use in hospitals has been approved by the European Society for Clinical Nutrition and Metabolism (ESPEN) and its validity has been confirmed through validation analyses in the literature [17]. The NRS-2002 score was determined based on the following parameters: impaired nutritional status (including BMI, recent changes in body weight, and food intake), severity of disease (including surgery, injuries, etc.), and age (patients over 70 years received an additional 1 point). An NRS-2002 score of ≥3 was considered indicative of nutritional risk.

### 2.3. Sarcopenia Risk Assessment

In the revised European consensus on the definition and diagnosis of sarcopenia (EWGSOP2), the SARC-F questionnaire was recommended for sarcopenia risk screening due to its high specificity and practical applicability [13]. The validity and reliability of the SARC-F have been confirmed in various healthcare settings [18]. The SARC-F (Simple Sarcopenia Screening Form) assesses parameters related to muscle mass and strength—such as muscle strength, assistance in walking, rising from a chair, climbing stairs, and history of falls—based on patients’ self-reported responses. Each item is scored on a Likert-type scale: 0 (none), 1 (some), and 2 (a lot or unable), with a total possible score of 10 points. A total score of 4 or higher indicates a risk of sarcopenia.

### 2.4. Statistical Analysis

The normality of continuous variables was tested using the Kolmogorov–Smirnov test. Data with a normal distribution were expressed as mean ± standard deviation (SD), whereas categorical variables were presented as counts and percentages. For univariate analyses, differences between groups were evaluated using Student’s *t*-test for continuous variables (when normality assumptions were satisfied) and the chi-square test for categorical variables. Variables showing statistical significance in univariate analyses were subsequently entered into a multivariate logistic regression model to identify independent predictors of malnutrition and sarcopenia risk. Model performance, as well as odds ratios (ORs) and 95% confidence intervals (CIs) for each variable, were calculated and reported. A two-tailed *p*-value < 0.05 was considered statistically significant for all analyses.

### 2.5. Ethics Committee Approval

Ethical approval for the study was obtained from the Dokuz Eylül University Non-Interventional Research Ethics Committee (Decision No: 2024/37-04, dated 6 November 2024). The study was conducted in accordance with the principles of the World Medical Association’s Declaration of Helsinki. In this cross-sectional clinical evaluation study, informed consent was obtained from all patients. Additionally, no assistance from artificial intelligence (AI) was used in the writing of this manuscript.

## 3. Results

### 3.1. Demographic Characteristics of the Patients

In this cross-sectional study, data from 1051 patients hospitalized with various diagnoses in 36 gastroenterology clinics across six different geographical and sociocultural regions of Turkey were analyzed. The types of healthcare institutions included university hospitals, training and research hospitals, and private hospitals. The mean age was 59.1 ± 16.9 years for women and 60.3 ± 18.2 years for men. In terms of gender distribution, women constituted 54.7% (575 patients) of the study population, while men accounted for 45.3% (476 patients). The most common reason for hospitalization was acute pancreatitis (17.4%), followed by choledocholithiasis (14.7%), liver cirrhosis (13.8%), upper gastrointestinal (GI) bleeding (13.1%), GI malignancy (9.0%), cholangitis (6.4%), ulcerative colitis (4.6%), lower GI bleeding (4.1%), and Crohn’s disease (3.4%). In addition to their primary gastrointestinal diseases, 38.4% of the patients had at least one comorbid condition, with a significant proportion presenting with multiple comorbidities. Among these, hypertension (12.1%), coronary artery disease (5.2%), diabetes mellitus (24.1%), chronic kidney disease (5.3%), and congestive heart failure (7.0%) were the most frequently observed. The mean body mass index (BMI) was 25.7 ± 5.5, and the median disease duration was 1 year (IQR: 0.1–6).

### 3.2. Prevalence of Malnutrition and Sarcopenia Risk in the Patient Population

In this study, population, the nutritional status of the patients was assessed using the NRS-2002 screening tool, and the prevalence of malnutrition risk was found to be 27.8%. Additionally, sarcopenia risk screening performed with the SARC-F questionnaire revealed that 32.7% of patients were at risk of sarcopenia at the time of hospitalization. The distribution of variables among patients with and without malnutrition risk, and those with and without sarcopenia risk, is summarized in Table 1 and Table 2.

In the study, the mean age of patients with sarcopenia was 69.6 ± 15.1 years, whereas it was 54.7 ± 16.5 years in those without sarcopenia (*p* < 0.001). Similarly, the mean age of patients with malnutrition risk was 67.6 ± 16.0 years, which was significantly higher compared to those without malnutrition risk (56.5 ± 17.1 years) (*p* < 0.001).

When examining the relationships between sarcopenia risk, malnutrition status, and various demographic and clinical variables, it was found that the prevalence of sarcopenia was 37.8% among women and 62.2% among men, and this difference was statistically significant (*p* = 0.001). However, the association between malnutrition status and gender was not statistically significant (*p* = 0.387).

In the analysis based on the type of healthcare institution where patients were hospitalized, no significant difference was found in malnutrition risk rates among university hospitals, training and research hospitals, and private hospitals (*p* = 0.076). The malnutrition risk rate was 30.4% in university hospitals, 24.0% in training and research hospitals, and 23.5% in private hospitals; however, these differences were not statistically significant.

When examining the relationship between sarcopenia risk and the type or regional distribution of healthcare institutions, the sarcopenia risk rate was 35.0% in university hospitals, 29.6% in training and research hospitals, and 23.5% in private hospitals. Nevertheless, these differences were not statistically significant (*p* = 0.136).

On the other hand, significant differences were observed in malnutrition risk rates across different regions (*p* = 0.007). The lowest malnutrition risk rate was found in the Black Sea region (14.1%), while the highest rate was observed in the Central Anatolia region (35.3%). Regarding the regional distribution of sarcopenia risk, the lowest rate was identified in the Black Sea region (27.2%), whereas the highest was in the Central Anatolia region (40.7%). In other regions, sarcopenia rates ranged between 30% and 40%. However, the differences among regions were not statistically significant (*p* = 0.279).

In the analysis based on admission diagnoses, patients diagnosed with acute pancreatitis had a significantly higher sarcopenia rate at 23.0% (*p* = 0.002), and the malnutrition risk rate was also notably higher at 21.9% (*p* = 0.049). This indicates a marked increase in the prevalence of sarcopenia and malnutrition among patients with pancreatitis. In inflammatory bowel diseases such as Crohn’s disease and ulcerative colitis, no significant association with malnutrition risk was observed (Crohn: *p* = 0.449; Ulcerative Colitis: *p* = 0.271). Among patients with cirrhosis, the sarcopenia rate was significantly high at 54.5% (*p* < 0.001), and malnutrition risk in the same group was also significantly elevated at 40.7% (*p* < 0.001). In patients with upper gastrointestinal (GI) bleeding, the risk of sarcopenia was significantly higher compared to those without (*p* = 0.013), whereas no significant association was found with malnutrition risk (*p* = 0.735). In patients diagnosed with GI malignancy, the sarcopenia risk rate was significantly high at 44.2% (*p* = 0.012), while the malnutrition risk rate was also notably elevated at 50.5%, indicating a strong association between these two clinical conditions (*p* < 0.001).

Among patients with diabetes mellitus, the sarcopenia risk rate was 42.7% (*p* < 0.001), and the malnutrition rate was 36.0% (*p* < 0.001). Diseases such as chronic pancreatitis, COPD, pneumonia, congestive heart failure, and chronic kidney disease were significantly associated with both sarcopenia and malnutrition. These results indicate that sarcopenia and malnutrition are particularly associated with primary gastrointestinal diseases such as cirrhosis and GIS malignancy, as well as with comorbid conditions such as diabetes mellitus, COPD, and chronic kidney disease. The management of these diseases may influence clinical practice, as they have a significant impact on both conditions.

When evaluated in terms of body mass index (BMI), no significant difference was found regarding the presence of sarcopenia risk (26.3 ± 6.3 vs. 26.4 ± 5.1, *p* = 0.732). In contrast, patients with malnutrition risk had significantly lower BMI values (24.7 ± 5.3) compared to those without malnutrition risk (27.1 ± 5.4) (*p* < 0.001).

In the study, the median disease duration was 1 (0.25–12) years in patients with sarcopenia risk, compared to 1 (0.1–4) years in those without sarcopenia risk, and this difference was statistically significant (*p* = 0.011). Similarly, the median disease duration in patients with malnutrition risk was 2 (1–12) years, whereas it was 1 (0.1–3) years in those without malnutrition risk, and this difference was also statistically significant (*p* < 0.001).

When examining the relationship between disease severity and the presence of sarcopenia and malnutrition risk, it was found that patients with gastrointestinal (GIS) malignancy who had sarcopenia and malnutrition risk had significantly higher ECOG scores compared to those without sarcopenia and malnutrition (*p* < 0.001). The upper GIS bleeding GBS score, however, was significantly associated with both sarcopenia and malnutrition risk, with higher scores observed in both groups (sarcopenia: *p* = 0.004; malnutrition: *p* = 0.041). The severity score for liver cirrhosis was also significantly associated with both sarcopenia and malnutrition risk (sarcopenia: *p* = 0.012; malnutrition: *p* = 0.015). In terms of Child classification, sarcopenia risk rates were higher in the Child B and Child C groups (*p* = 0.010). However, since malnutrition risk rates were also high in the Child A group, no statistically significant difference was observed among the Child classes (*p* = 0.052). This also indicates that both conditions are observed more frequently as disease severity increases. The Ulcerative Colitis Mayo score showed a significant association with sarcopenia risk (*p* = 0.017), but no such association was found with malnutrition risk(*p* = 0.489). The Acute Pancreatitis BISAP score demonstrated a strong association with both conditions (sarcopenia: *p* < 0.001; malnutrition: *p* < 0.001) (Table 3).

According to the results of the Pearson Correlation Analysis, significant relationships were observed between sarcopenia and malnutrition risk screening and various clinical variables. A strong positive correlation was found between the sarcopenia assessment (SARC-F) and the Malnutrition Risk Screening (NRS-2002) (r = 0.544, *p* < 0.001), indicating that sarcopenia and malnutrition risks are interrelated conditions. Significant positive correlations were also found between age and both variables (sarcopenia: r = 0.447, *p* < 0.001; malnutrition: r = 0.387, *p* < 0.001). As age increases, both conditions become more prevalent. A strong relationship was found between the GIS malignancy ECOG score and sarcopenia (r = 0.603, *p* < 0.001), and a similarly strong correlation was observed with malnutrition risk (r = 0.646, *p* < 0.001). In addition, significant correlations were found between the acute pancreatitis BISAP score and sarcopenia risk (r = 0.406, *p* < 0.001) as well as malnutrition risk (r = 0.509, *p* < 0.001), revealing that both conditions are more prevalent in patients with pancreatitis. Positive correlations were also observed between the Child score in liver cirrhosis and sarcopenia (r = 0.35, *p* < 0.001) and malnutrition risk (r = 0.254, *p* = 0.014), indicating that as the severity of cirrhosis increases, both conditions become more prominent (Table 4).

The logistic regression analysis on factors associated with sarcopenia risk demonstrates that various variables significantly affect the risk of sarcopenia. Age increases the risk of sarcopenia by 5% for each unit increase, and this relationship is statistically significant (OR = 1.05, 95% CI: [1.04–1.06], *p* < 0.001). Male gender appears as a protective factor, reducing the risk of sarcopenia risk (OR = 0.55, 95% CI: [0.40–0.75], *p* < 0.001). Malnutrition is a strong risk factor for sarcopenia, with individuals who are malnourished having approximately a fivefold higher risk (OR = 5.16, 95% CI: [3.70–7.21], *p* < 0.001). Liver cirrhosis is another important risk factor for sarcopenia (OR = 2.24, 95% CI: [1.47–3.42], *p* < 0.001). Additionally, heart failure shows a positive association with sarcopenia risk (OR = 1.99, 95% CI: [1.13–3.50], *p* = 0.016).

Factors associated with malnutrition risk also show significant relationships with various variables. Age is again an important determinant of malnutrition risk; with each additional year, the risk of malnutrition increases by 3% (OR = 1.03, 95% CI: [1.02–1.04], *p* < 0.001). Male gender is positively associated with malnutrition risk, with men having a higher risk of malnutrition (OR = 1.39, 95% CI: [1.02–1.91], *p* = 0.039). Malignancy is significantly associated with malnutrition risk, with the presence of malignancy increasing the risk by 2.2-fold (OR = 2.2, 95% CI: [1.27–3.83], *p* = 0.005) (Table 5).

These results indicate that the risks of sarcopenia and malnutrition are interrelated conditions and are significantly associated with clinical factors such as age, gender, liver cirrhosis, heart failure, malignancy, and choledocholithiasis.

## 4. Discussion

In this multicenter cross-sectional study, the nutritional and sarcopenia risks of 1051 patients hospitalized in gastroenterology clinics across different regions of Türkiye were evaluated using the NRS-2002 and SARC-F questionnaires. The prevalence of malnutrition risk was 27.8% and sarcopenia risk was 32.7%, consistent with previous studies showing these conditions are common among hospitalized patients. Independent factors associated with malnutrition risk included advanced age, male sex, presence of sarcopenia, and malignancy, whereas sarcopenia risk was related to advanced age, female sex, malnutrition, liver cirrhosis, and heart failure. Unlike other studies in the literature, this study not only evaluated the nutritional status of the patients but also identified the risk of sarcopenia and the factors that may contribute to the development of both.

According to the results of our study, nearly one-third of the patients admitted to gastroenterology clinics were malnourished at the time of hospitalization. In a previous study conducted in four hospitals in England, the malnutrition rate at admission was found to be 20%, and in malnourished patients, the length of hospital stay was prolonged, new infections developed, and disease severity increased [19]. The higher malnutrition rate in our study may be due to the fact that the screening was conducted only among patients hospitalized in gastroenterology clinics, and unlike that study, the validated NRS-2002 questionnaire was used instead of BMI. In addition, a malnutrition screening study conducted in approximately 10,000 patients during Nutrition Day in the United States using the Malnutrition Screening Tool (MST) found a malnutrition rate of 32.1% at hospital admission, similar to our findings [20]. Recently, Jiang et al. evaluated the nutritional risk of 15,098 patients hospitalized in six departments (neurology, gastroenterology, nephrology, pulmonology, general surgery, and thoracic surgery) across 13 cities and 19 major hospitals in China within 72 h of admission using the NRS-2002. They found a malnutrition prevalence of 12% and a malnutrition risk rate of 35.5%, which is similar to our findings [21]. In a multinational study involving 5051 patients from 12 European countries and several Middle Eastern countries, the malnutrition risk among hospitalized patients was reported as 32.6% [11]. The differences in risk rates across these studies likely stem from variations in patient populations due to differences in countries, hospitals, and clinical settings, as well as the use of different methods for malnutrition screening. Based on studies conducted in Europe and the United States, it is estimated that approximately 31% of all hospitalized patients are malnourished or at risk of malnutrition [22].

In Turkey, a multicenter study conducted by Korfalı et al. in 2009 evaluated a total of 29,139 patients hospitalized in various clinics across 34 hospitals in 19 cities [12]. Using the NRS-2002 screening tool, it was determined that 15% of these patients were at nutritional risk at the time of admission. In the subgroup analysis of this study, 1677 patients hospitalized in gastroenterology clinics were included, and the prevalence of malnutrition risk among these patients was found to be 19.1% [12]. In a study conducted in a surgical clinic in Bursa, Gündoğdu et al. detected malnutrition in 33.3% of hospitalized patients using the Subjective Global Assessment (SGA) [23]. Similarly, Sungurtekin et al., when evaluating 251 patients admitted to internal medicine and surgical clinics at a university hospital in Denizli, found a malnutrition rate of 30% using SGA and 36% when using the Nutrition Risk Index (NRI) [24]. In a study conducted by Kuzu et al. in the surgical clinic of a university hospital in Ankara, 460 patients admitted for major elective surgery were evaluated, and the prevalence of malnutrition was reported as 58.3% using the Nutrition Risk Index (NRI), 63.5% using the Maastricht Index (MI), and 67.4% using the Subjective Global Assessment (SGA) [25]. In summary, previous studies conducted in our country have generally been single-center studies with small patient populations, and no study has specifically evaluated patients hospitalized in gastroenterology clinics. Due to the use of different assessment methods in these studies, substantial variations in malnutrition rates have been observed. In our multicenter study, we preferred the NRS-2002 questionnaire to screen for malnutrition risk, as it is a validated tool and increasingly recognized as a new standard method for nutritional risk classification.

In this multicenter study, using data obtained from six different geographical regions of Türkiye and three different types of healthcare institutions, it was demonstrated that the prevalence of sarcopenia risk and malnutrition risk may vary according to both regions and types of institutions. Although the prevalence of sarcopenia risk did not show a statistically significant difference among regions, the highest rate was observed in the Central Anatolia Region (40.7%) and the lowest in the Black Sea Region (27.2%). This finding suggests that sarcopenia cannot be explained solely by regional socioeconomic differences, and that individual risk factors as well as region-specific lifestyle characteristics may also play a role. However, the lack of a statistically significant difference may be attributed to the imbalance in sample sizes. In terms of malnutrition risk, significant differences were observed among geographical regions. Notably, the rates of malnutrition risk were markedly higher in the Central Anatolia (35.3%) and Southeastern Anatolia (33.9%) regions compared to other areas. This finding indicates that regional variations in economic development, dietary habits, and access to healthcare services may have potential effects on malnutrition. When evaluated by type of healthcare institution, both sarcopenia and malnutrition risk rates were found to be higher in university hospitals. This is likely due to the fact that patients admitted to university hospitals are generally older, have more complex medical conditions, and carry a higher burden of chronic diseases.

Many risk factors such as low socioeconomic status, advanced age, severe illness, length of hospital stay, and lack of medical awareness are responsible for the development of malnutrition [26]. Advanced age is correlated with an increased risk of developing malnutrition, with the risk rising by 4% per year of age [27]. In hospitalized elderly patients, screenings performed using the Mini Nutritional Assessment (MNA) have shown that the prevalence of malnutrition can reach up to 22%. Prevalence rates are particularly higher among women, individuals over 80 years of age, and those with chronic diseases. Of course, there are significant geographical differences in the prevalence of malnutrition among the elderly. For example, the rates are relatively lower in Europe (2.1%) and Asia (4.8%), but higher in countries such as Iran (12.2%), India (16.3%), and Ethiopia (26.6%) [28]. Elderly individuals are at a higher risk of malnutrition due to various age-related factors that affect nutritional status, such as lack of physical activity, loss of appetite, decreased food intake, progressive decline in functional autonomy, comorbid diseases, economic and social isolation, and feelings of neglect. In our study population, both sarcopenia and malnutrition risk were found to increase with advancing age.

Compared to men, women face a higher risk of malnutrition due to various factors such as longer life expectancy and a greater likelihood of experiencing adverse economic and social conditions in old age. However, in our study, regression analyses demonstrated the opposite, showing that male gender was a risk factor for malnutrition. This discrepancy may be attributed to the heterogeneity of the study population, the higher mean age among men, as well as differences in BMI and sociocultural characteristics. Additionally, the decline in self-care ability observed in men with advancing age may also contribute to this finding. Social factors such as low educational level and living alone also play a significant role in the risk of malnutrition [29]. In this multicenter study, BMI was found to be lower in patients with malnutrition risk compared to those without.

Screening and treatment of malnutrition in patients with gastrointestinal diseases are important, as they influence clinical outcomes of the disease. For example, in a study conducted in Canada including 1015 patients from 18 hospitals, 30.4% of patients had gastrointestinal diseases, and 45% were found to be malnourished [30]. In the present study, malnutrition risks of patients hospitalized in the gastroenterology department were also screened, and it was found that patients with liver cirrhosis and gastrointestinal malignancies had significantly higher risks of malnutrition and sarcopenia. When the impact of disease severity on malnutrition risk at hospital admission was examined, it was demonstrated that in patients admitted with diagnoses such as upper gastrointestinal bleeding, liver cirrhosis, acute pancreatitis, and gastrointestinal malignancies, malnutrition risk increased in parallel with disease severity. Subgroup analyses further revealed that in patients hospitalized with liver cirrhosis, the likelihood of malnutrition risk increased as the Child score increased. Moreover, disease duration was also shown to have a significant effect on the risk of malnutrition and sarcopenia.

Cancer patients are at high risk of malnutrition due to both the disease itself and the cancer treatments they receive, which affect their nutritional status. Therefore, all cancer patients should be screened for malnutrition risk at an early stage, regardless of BMI [31]. It is estimated that in 10–20% of cancer patients, death occurs due to disease-related malnutrition rather than the cancer itself [32]. While Waitzberg et al. reported a threefold higher rate of malnutrition in cancer patients compared to other patient groups, Planas et al. found similar malnutrition prevalence rates between cancer patients and those with other diseases [1,33]. A European study showed that in 40% of cancer patients, physicians overlooked the severity of cancer-related malnutrition, resulting in many severely malnourished cancer patients not receiving adequate nutritional support [34]. In our study, the risk of malnutrition in patients hospitalized with a diagnosis of gastrointestinal malignancy was 50.5%, higher than in patients with other diseases. In cancer patients, malnutrition risk increases due to age, inadequate nutrition, tumor type, disease stage, and treatment-related side effects [35]. The ESPEN oncology expert group also recommends that the nutritional status of all cancer patients be screened early in their treatment, signs and symptoms of anorexia, cachexia, and, if possible, sarcopenia be identified, nutritional and metabolic support be provided as part of cancer therapy, and physical activity be promoted [31].

In our study, malnutrition risk was detected in 29% of patients hospitalized with a diagnosis of upper GI bleeding, and no statistical difference was found compared to those without malnutrition risk. However, a correlation was observed between the increase in malnutrition risk and the severity of the disease assessed by the Glasgow-Blatchford score. In a recent study conducted by Jaan and colleagues using the American national dataset, it was reported that between 2016 and 2020, 742,592 patients were hospitalized with non-variceal upper gastrointestinal bleeding, with a malnutrition rate of 10.32% and a severe malnutrition rate of 4%. The study also highlighted that malnutrition was more prevalent among elderly female patients [27]. In our study, the higher risk of malnutrition among patients hospitalized with upper gastrointestinal bleeding may be attributed to the higher mean age and comorbidity rates of our patient population compared to those in the aforementioned study.

In liver cirrhosis, increased catabolic processes disrupt the balance between energy demand and intake. Complications of portal hypertension (such as ascites and hepatic encephalopathy) contribute to the development of malnutrition by causing anorexia, reduced physical activity, early satiety, and an increase in basal metabolic rate. Additionally, systemic inflammation, endocrine factors, and metabolic dysregulation are also responsible for the development of malnutrition in liver cirrhosis [36]. In our study, malnutrition risk was identified in a high proportion of 40.7% among patients diagnosed with liver cirrhosis who were hospitalized in gastroenterology clinics. Furthermore, in correlation with disease severity, the malnutrition risk rate was found to be 30.4% in Child A cirrhosis and 53.3% in Child C cirrhosis.

Early diagnosis of malnutrition and the provision of adequate nutritional support are among the fundamental components of treatment in acute pancreatitis (AP), chronic pancreatitis (CP), and pancreatic cancer (PC). In CP, two main mechanisms are responsible for the development of malnutrition. The first is the reduction in food digestion due to decreased enzyme secretion resulting from pancreatic exocrine insufficiency. The second is patients’ avoidance of eating and inadequate caloric intake due to postprandial abdominal pain. Moreover, smoking and alcohol consumption further deteriorate nutritional status [37]. While the prevalence of malnutrition in the context of CP has been reported to be around 30% in the literature [38], it was identified in approximately 70% of the relatively small number of CP patients in our study. In patients with AP, malnutrition was shown to correlate with an increase in the BISAP disease severity score, and the rate of malnutrition among those hospitalized with a diagnosis of AP was found to be 21%.

In inflammatory bowel disease (IBD), malnutrition and low serum micronutrient levels negatively affect both induction and maintenance of remission, thereby also impairing patients’ quality of life [39]. The higher prevalence and severity of malnutrition observed in Crohn’s disease (CD) compared to ulcerative colitis (UC) can be attributed to the more extensive involvement of the main sites of nutrient absorption in CD, extensive mucosal involvement, fistula formation, the development of short bowel syndrome after resection, and gastrointestinal tract obstruction [40]. In our patient population, the rate of malnutrition risk was found to be 20.8% among those hospitalized with UC and 33.3% among those with CD. The lack of a statistically significant correlation between malnutrition and other variables in our analysis may be explained by the fact that these patients receive adequate nutritional support during their follow-up outside the hospital.

In diseases such as chronic obstructive pulmonary disease (COPD), congestive heart failure (CHF), and chronic kidney disease (CKD), the catabolic process is triggered due to the increase in chronic inflammatory cytokines, leading to the development of malnutrition. The persistence of inflammation mediated by proinflammatory cytokines affects neural centers, resulting in loss of appetite, increased skeletal muscle metabolism, delayed gastric emptying, and dysregulation of appetite hormones [28]. Furthermore, heart failure and diabetes mellitus (DM) are diseases associated with high rates of malnutrition and in-hospital mortality. In our study, malnutrition was found to be more frequent in patients with comorbid conditions such as motility disorders, DM, chronic pancreatitis (CP), COPD, and CKD, whereas no increased risk was demonstrated in patients with CHF. This may be attributed to the small number of patients diagnosed with CHF in our study population. However, Carrero et al. demonstrated a prevalence of protein–energy malnutrition ranging from 11% to 54% in patients with stage 3–5 chronic kidney disease (CKD) [41]. In the present study, when evaluated in terms of comorbidities other than primary gastrointestinal diseases, malnutrition was identified in 23.6% of patients with coronary artery disease (CAD), 42.7% with diabetes mellitus (DM), 47.9% with chronic obstructive pulmonary disease (COPD), 36.5% with congestive heart failure (CHF), and 53.6% with chronic kidney disease (CKD).

There is a significant association between malnutrition and sarcopenia. Protein–energy malnutrition, in particular, is one of the main causes of sarcopenia and is frequently observed in the elderly [42]. In this context, the demonstrated correlation between sarcopenia risk and malnutrition risk in the present study further supports this relationship. In a multicenter study conducted in Turkey in 2020, sarcopenia risk was identified in 48.8% of 492 hospitalized patients using the SARC-F questionnaire. The risk of sarcopenia was found to be higher, particularly among women and older patients [16]. The higher rate of sarcopenia reported in that study compared to our findings may be attributed to the fact that the study population consisted exclusively of patients aged over 65 years. In our study, the risk of sarcopenia was identified as 37.8% in women and 28.5% in men. Although skeletal muscle loss begins at the age of 35, it occurs at a rate of 1–2% per year. After the age of 65, this muscle loss increases to 3% per year [43]. Stress-related inflammatory cytokines, elevated cortisol levels, and acute illnesses contribute to both malnutrition and the development or exacerbation of sarcopenia due to limited mobilization during hospitalization [16]. In this study, age and female sex were independently associated with sarcopenia risk. Lee et al. reported that age is the most significant variable associated with sarcopenia [44]. Age is also a component of the Ishii formula, which provides a practical method for defining sarcopenia. As observed in many geriatric syndromes, female patients are at a higher risk of developing sarcopenia. The lower prevalence of sarcopenia in men can be attributed, in particular, to the more rapid hormonal changes affecting skeletal muscle loss in women. After the age of 65, the decline in sex hormones such as estrogen and androgens is more pronounced in women [43]. Additionally, frailty and the scales determining SARC-F incorporate muscle strength, which is generally lower in women. Furthermore, with advancing age, the decline in muscle strength occurs earlier than the loss of muscle mass, and this phenomenon is more pronounced in women [45].

In our study, among patients hospitalized with liver cirrhosis, the risk of sarcopenia assessed using the SARC-F questionnaire was 54.5%, with a rate of 9.2% in Child A patients and 52.6% in Child C patients. In the evaluation conducted by Majeed et al. in 416 cirrhotic patients, the prevalence of sarcopenia was 10.09%; however, the presence of sarcopenia was associated with increases in Child and MELD scores, history of upper gastrointestinal bleeding, ascites, and hepatic encephalopathy. No differences were identified with respect to cirrhosis etiology [46]. While sarcopenia was diagnosed using CT in that study, the risk of sarcopenia in our study was determined via a questionnaire, which explains the discrepancy in rates between the two studies. The unknown etiology of cirrhosis in our patient population may suggest that a potential predominance of alcohol-related etiology could account for the higher sarcopenia risk observed. Nevertheless, both the risk and presence of sarcopenia increase in parallel with the severity of the underlying liver disease.

Sarcopenia can be observed in up to 60% of patients with chronic pancreatitis (CP), whereas data regarding acute pancreatitis (AP) remain unclear [47]. In our study, the risk of sarcopenia was 23% in patients with AP and 57.1% in those with CP. Similarly, in an evaluation conducted by Ramsey et al. involving 49 AP and 54 CP patients using cross-sectional imaging, the prevalence of sarcopenia was higher in CP compared to AP, consistent with our findings (83.3% vs. 46.9%) [48].

In IBD, in addition to malnutrition, chronic inflammation, increased inflammatory activity within adipose tissue, vitamin deficiencies, and disruptions in the muscle–gastrointestinal axis are responsible for the development of sarcopenia [39]. A recent meta-analysis demonstrated that the prevalence of sarcopenia was 52% in Crohn’s disease (CD) and 37% in ulcerative colitis (UC), indicating that sarcopenia represents a significant clinical problem in this patient population [49]. In cases of acute severe ulcerative colitis (UC), sarcopenia rates have been reported to reach up to 70% [50]. Similarly, in a 2021 study conducted in Turkey by Ünal et al. involving 344 patients and based on the European Working Group on Sarcopenia in Older People 2 (EWGSOP2) criteria, the prevalence of sarcopenia in patients with inflammatory bowel disease (IBD) was found to be 41.3% [51]. In contrast to the literature, in our study, the risk of sarcopenia was identified as 11.1% in Crohn’s disease (CD) and 20.8% in UC. This discrepancy may be attributed to the smaller number of IBD patients in our study, the lower mean age of the study population, or the possibility that these patients were already receiving adequate nutritional support. Moreover, studies using sarcopenia screening tools such as SARC-F in IBD patients are limited. Additionally, variations in diagnostic criteria for sarcopenia and differences in body composition related to ethnic background may also explain the lower rates observed in our findings.

Sarcopenia is also common in chronic comorbid conditions such as cardiovascular diseases, chronic kidney failure, and cancer. In elderly patients in particular, it is associated with rapid progression of cardiovascular disease, higher mortality rates, and decreased quality of life. Although the underlying pathophysiological mechanism is complex, the imbalance between anabolic and catabolic muscle homeostasis—independent of neuronal degeneration—plays a predominant role [52]. Insulin resistance in type 2 diabetes mellitus (T2DM) increases the risk of sarcopenia threefold. Sarcopenia is particularly associated with microvascular and macrovascular complications in T2DM. Moreover, sarcopenia has been reported in 8.2% of newly diagnosed T2DM patients [53]. In another study, the prevalence of sarcopenia was found to be 15% among patients with stable chronic obstructive pulmonary disease (COPD) [54].

This study has several limitations that should be considered when interpreting the findings. First, its cross-sectional design prevents the establishment of causal relationships between malnutrition, sarcopenia, and the associated clinical factors. Second, data were collected during a single week, which may not capture seasonal or temporal variations in patient characteristics or nutritional status. Third, the study population consisted solely of individuals hospitalized in gastroenterology clinics, which may limit the generalizability of the results to broader outpatient or community settings. The heterogeneity of the underlying gastrointestinal and systemic diseases may also have acted as a confounding factor, despite the adjustments made in the regression analyses. Finally, potential inter-clinic variability across the 36 participating centers and unmeasured factors such as socioeconomic status, medication use, and physical activity may have influenced the observed associations. Additionally, SARC-F has been shown to be more sensitive in selected patient populations, such as older individuals with a higher number of comorbidities and lower levels of physical activity. However, due to its high specificity, SARC-F can effectively exclude non-sarcopenic patients and may therefore be a useful tool for screening sarcopenia risk in hospitalized patients. Nevertheless, its low sensitivity may lead to the omission of patients with less severe sarcopenia. The use of the self-reported SARC-F questionnaire may introduce a degree of reporting bias, as patients’ subjective responses could affect the accuracy of sarcopenia risk assessment. Although the study included a large and nationally representative inpatient cohort, minor variations in data collection across centers could not be entirely eliminated.

In conclusion, this study is the first in Turkey to examine the risk of malnutrition and sarcopenia, along with the factors influencing them, in patients hospitalized in gastroenterology clinics. Nearly one-third of the patients admitted to gastroenterology clinics were found to be at risk for malnutrition and sarcopenia. Furthermore, this study demonstrates that sarcopenia and malnutrition may vary regionally and institutionally across Turkey. In addition, advanced age, male gender, presence of malignancy, and coexisting sarcopenia were identified as independent risk factors for malnutrition. Similarly, advanced age, female gender, presence of malnutrition, diagnosis of liver cirrhosis, and coexisting heart failure were identified as risk factors for sarcopenia. Although malnutrition is common, it is often unrecognized or underestimated. Many physicians are unaware of their patients’ nutritional status. Therefore, awareness on this issue should be increased, and it should be adopted as a health policy. Healthcare professionals in hospitals should first be trained in the early screening of malnutrition and should also treat patients in whom malnutrition is detected.

## Figures and Tables

**Table 1 diagnostics-15-02935-t001:** Malnutrition and sarcopenia risk status by type of healthcare institution and region.

Category	Subheading	Total (*n*, %)	No Malnutrition Risk (*n*, %)	Malnutrition Risk Present (*n*, %)	*p* Value	No Sarcopenia Risk (*n*, %)	Sarcopenia Risk Present (*n*, %)	*p*
Type of Healthcare Institution								
	University Hospital	622 (59.2%)	433 (69.6%)	189 (30.4%)		404 (65.0%)	218 (35.0%)	
	Training and Research Hospital	412 (39.2%)	313 (76.0%)	99 (24.0%)		290 (70.4%)	122 (29.6%)	
	Private Hospital	17 (1.6%)	13 (76.5%)	4 (23.5%)		13 (76.5%)	4 (23.5%)	
	Total	1051(100%)	759 (72.2%)	292 (27.8%)	0.076	707 (67.3%)	344 (32.7%)	0.136
Region								
	Aegean	276 (26.4%)	199 (72.1%)	77 (27.9%)		183 (66.3%)	93 (33.7%)	
	Marmara	276 (26.4%)	206 (74.6%)	70 (25.4%)		190 (68.8%)	86 (31.2%)	
	Mediterranean	129 (12.3%)	92 (71.3%)	37 (28.7%)		88 (68.2%)	41 (31.8%)	
	Black Sea	92 (8.8%)	79 (85.9%)	13 (14.1%)		67 (72.8%)	25 (27.2%)	
	Central Anatolia	150 (14.3%)	97 (64.7%)	53 (35.3%)		89 (59.3%)	61 (40.7%)	
	Southeastern Anatolia	124 (11.8%)	82 (66.1%)	42 (33.9%)		86 (69.4%)	38 (30.6%)	
	Total	1047(100%)	755 (72.1%)	292 (27.9%)	0.007	703 (67.1%)	344 (32.9%)	0.279

**Table 2 diagnostics-15-02935-t002:** Distribution of sarcopenia and malnutrition risk across all variables.

Variable	No Sarcopenia Risk (*n*, %)	Sarcopenia Risk Present (*n*, %)	*p*	No Malnutrition Risk (*n*, %)	Malnutrition Risk Present (*n*, %)	*p*
Age, mean (SD), years	54.7 (16.5)	69.6 (15.1)	<0.001	56.5 (17.1)	67.6 (16.0)	<0.001
Age, *n* (%)			<0.001			<0.001
<60	390 (85.5)	66 (14.5)		388 (85.1)	68 (14.9)	
60 and above	317 (53.3)	278 (46.7)		371 (62.4)	224 (37.6)	
Gender			0.001			0.387
Female	296 (62.2%)	180 (37.8%)		350 (73.5%)	126 (26.5%)	
Male	411 (71.5%)	164 (28.5%)		409 (71.1%)	166 (28.9%)	
**Admission Diagnoses**						
Acute Pancreatitis	141 (77.0%)	42 (23.0%)	0.002	143 (78.1%)	40 (21.9%)	0.049
Ulcerative Colitis	38 (79.2%)	10 (20.8%)	0.072	38 (79.2%)	10 (20.8%)	0.271
Crohn’s Disease	32 (88.9%)	4 (11.1%)	0.005	24 (66.7%)	12 (33.3%)	0.449
Cirrhosis	66 (45.5%)	79 (54.5%)	<0.001	86 (59.3%)	59 (40.7%)	<0.001
Upper GI Bleeding	80 (58.0%)	58 (42.0%)	0.013	98 (71.0%)	40 (29.0%)	0.735
Lower GI Bleeding	24 (55.8%)	19 (44.2%)	0.102	29 (67.4%)	14 (32.6%)	0.475
GI Malignancy	53 (55.8%)	42 (44.2%)	0.012	47 (49.5%)	48 (50.5%)	<0.001
Cholangitis	38 (56.7%)	29 (43,3%)	0.057	43 (64.2%)	24 (35.8%)	0.129
Choledocholithiasis	126 (81.3%)	29 (18.7%)	<0.001	136 (87.7%)	19 (12.3%)	<0.001
Others	110 (74.3%)	38 (25.7%)	0.048	120 (81.1%)	28 (18.9%)	0.009
**Comorbid Diseases**	374 (57.8%)	273 (42.2%)	<0.001	426 (65.8%)	221 (34.2%)	<0.001
Atrial Fibrillation	6 (46.2%)	7 (53.8%)	0.103	7 (53.8%)	6 (46.2%)	0.208
Hypothyroidism	6 (75.0%)	2 (25.0%)	0.640	7 (87.5%)	1 (12.5%)	0.333
Hyperlipidemia	9 (81.8%)	2 (18.2%)	0.301	10 (90.9%)	1 (9.1%)	0.164
Hypertension	81 (63.8%)	46 (36.2%)	0.371	100 (78.7%)	27 (21.3%)	0.080
Coronary Artery Disease	30 (54.5%)	25 (45.5%)	0.039	42 (76.4%)	13 (23.6%)	0.481
Other/Motility Disorders	91 (54.5%)	76 (45.5%)	<0.001	108 (64.7%)	59 (35.3%)	0.018
History of Surgery	30 (65.2%)	16 (34.8%)	0.762	30 (65.2%)	16 (34.8%)	0.278
Psychiatric Disorder	5 (41.7%)	7 (58.3%)	0.057	7 (58.3%)	5 (41.7%)	0.280
Hyperthyroidism	5 (50.0%)	5 (50.0%)	0.242	5 (50.0%)	5 (50.0%)	0.115
Diabetes Mellitus	145 (57.3%)	108 (42.7%)	<0.001	162 (64.0%)	91 (36.0%)	<0.001
Chronic Pancreatitis	3 (42.9%)	4 (57.1%)	0.167	2 (28.6%)	5 (71.4%)	0.010
COPD (Chronic Obstructive Pulmonary Disease)	25 (52.1%)	23 (47.9%)	0.022	27 (56.3%)	21 (43.8%)	0.011
Pneumonia	4 (57.1%)	3 (42.9%)	0.567	5 (71.4%)	2 (28.6%)	0.963
Congestive Heart Failure	30 (40.5%)	44 (59.5%)	<0.001	47 (63.5%)	27 (36.5%)	0.083
Chronic Kidney Disease	28 (50.0%)	28 (50.0%)	0.005	26 (46.4%)	30 (53.6%)	<0.001
Presence of Malignancy	34 (48.6%)	36 (51.4%)	<0.001	34 (48.6%)	36 (51.4%)	<0.001
Pseudocyst/WON (Walled-off Necrosis)	5 (100.0%)	0 (0.0%)	0.118	1 (20.0%)	4 (80.0%)	0.009
CVA–DVT (Cerebrovascular Accident–Deep Vein Thrombosis)	4 (57.1%)	3 (42.9%)	0.567	6 (85.7%)	1 (14.3%)	0.424
BMI, mean (SD)	26.4 (5.1)	26.3 (6.3)	0.732	27.1 (5.4)	24.7 (5.3)	<0.001
Disease duration, median (IQR)	1 (0.1–4)	1 (0.25–12)	0.011	1 (0.1–3)	2 (1–12)	<0.001

Categorical variables were analyzed using the chi-square test, and continuous variables were analyzed with Student’s *t*-test or the Mann–Whitney U test, as appropriate.

**Table 3 diagnostics-15-02935-t003:** Relationship between disease severity, malnutrition, and sarcopenia.

Variable	No Sarcopenia Risk (*n*, %)	Sarcopenia Risk Present (*n*, %)	*p*	No Malnutrition Risk (*n*, %)	Malnutrition Risk Present (*n*, %)	*p*
GIS Malignancy ECOG Score (Mean ± SD)	1.06 ± 0.98	2.08 ± 1.08	<0.001	0.91 ± 0.86	2.14 ± 1.05	<0.001
Lower GIS Bleeding Oakland Score (Mean ± SD)	16.30 ± 10.28	16.18 ± 8.03	0.966	16.52 ± 9.66	15.69 ± 8.78	0.796
Upper GIS Bleeding GBS Score (Mean ± SD)	8.45 ± 4.51	10.64 ± 3.55	0.004	8.87 ± 4.35	10.57 ± 3.86	0.041
Liver Cirrhosis Severity Score (Mean ± SD)	7.76 ± 3.11	9.31 ± 2.74	0.012	8.00 ± 2.93	9.54 ± 2.90	0.015
Ulcerative Colitis Mayo Score (Mean ± SD)	3.89 ± 2.60	6.63 ± 3.62	0.017	4.26 ± 3.17	4.89 ± 2.09	0.489
Acute Pancreatitis BISAP Score (Mean ± SD)	0.93 ± 1.00	1.68 ± 1.19	<0.001	0.85 ± 0.95	2.00 ± 1.10	<0.001
Liver Cirrhosis Severity (Child)						
Child A	16 (25.4%)	7 (9.2%)		16 (69.6%)	7 (30.4%)	
Child B	27 (42.9%)	29 (38.2%)	0.010	37 (66.1%)	19 (33.9%)	0.052
Child C	20 (31.7%)	40 (52.6%)		28 (46.7%)	32 (53.3%)	

Categorical variables were analyzed using the chi-square test, and continuous variables were analyzed with Student’s *t*-test.

**Table 4 diagnostics-15-02935-t004:** Pearson correlation analysis between variables.

Variable 1	Variable 2	Pearson Correlation Coefficient (r)	*p*-Value
Sarcopenia Assessment (SARC-F)	Malnutrition Risk Screening (NRS-2002)	0.544	<0.001
Sarcopenia Assessment (SARC-F)	Age	0.447	<0.001
Sarcopenia Assessment (SARC-F)	Disease Duration (months)	0.017	0.589
Sarcopenia Assessment (SARC-F)	Cholangitis Severity (Tokyo)	0.137	0.503
Sarcopenia Assessment (SARC-F)	GIS Malignancy Performance (ECOG)	0.603	<0.001
Sarcopenia Assessment (SARC-F)	Lower GIS Bleeding Severity (Oakland)	0.091	0.577
Sarcopenia Assessment (SARC-F)	Upper GIS Bleeding Severity (GBS)	0.315	<0.001
Sarcopenia Assessment (SARC-F)	Liver Cirrhosis Severity (Numeric)	0.35	<0.001
Sarcopenia Assessment (SARC-F)	Ulcerative Colitis Severity (Mayo)	0.373	0.014
Sarcopenia Assessment (SARC-F)	Acute Pancreatitis Severity (BISAP)	0.406	<0.001
Malnutrition Risk Screening (NRS-2002)	Age	0.387	<0.001
Malnutrition Risk Screening (NRS-2002)	Disease Duration (months)	−0.007	0.829
Malnutrition Risk Screening (NRS-2002)	Cholangitis Severity (Tokyo)	−0.186	0.363
Malnutrition Risk Screening (NRS-2002)	GIS Malignancy Performance (ECOG)	0.646	<0.001
Malnutrition Risk Screening (NRS-2002)	Lower GIS Bleeding Severity (Oakland)	0.095	0.561
Malnutrition Risk Screening (NRS-2002)	Upper GIS Bleeding Severity (GBS)	0.275	0.002
Malnutrition Risk Screening (NRS-2002)	Liver Cirrhosis Severity (Numeric)	0.254	0.014
Malnutrition Risk Screening (NRS-2002)	Ulcerative Colitis Severity (Mayo)	0.084	0.594
Malnutrition Risk Screening (NRS-2002)	Acute Pancreatitis Severity (BISAP)	0.509	<0.001

**Table 5 diagnostics-15-02935-t005:** Logistic regression analysis results.

**Factors Associated with Sarcopenia Risk**			
**Variable**	**OR (Exp(B))**	**95% CI (OR)**	***p* Value**
Age	1.05	[1.04–1.06]	<0.001
Male sex (Reference: Female)	0.55	[0.40–0.75]	<0.001
Malnutrition	5.16	[3.70–7.21]	<0.001
Liver cirrhosis	2.24	[1.47–3.42]	<0.001
Choledocholithiasis	0.48	[0.29–0.78]	0.003
Heart failure	1.99	[1.13–3.50]	0.016
**Factors Associated with Malnutrition**			
**Variable**	**OR (Exp(B))**	**95% CI (OR)**	***p* Value**
Age	1.03	[1.02–1.04]	<0.001
Male sex (Reference: Female)	1.39	[1.02–1.91]	0.039
Sarcopenia	5.04	[3.63–7.01]	<0.001
Malignancy	2.20	[1.27–3.83]	0.005
Choledocholithiasis	0.36	[0.21–0.62]	<0.001

## Data Availability

The original contributions presented in this study are included in the article. Further inquiries can be directed to the corresponding authors.

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
