# Peer review of "Prevalence and Clinical Associations of Malnutrition and Sarcopenia Risk in Gastroenterology Inpatients: A Multicenter Cross-Sectional Study in Turkey"

_diagnostics, 2025, doi:10.3390/diagnostics15222935_

Round 1

Reviewer 1 Report

Comments and Suggestions for Authors

In their manuscript, authors raised very important issue which is prevalence risk of malnutrition and sarcopenia among in-patients. In this study authors choose heterogenous group of  1 047 patients hospitalized in the 36 gastroenterology clinics across 22 cities representing different geographical regions of Turkey. Authors used validated, widely used malnutrition and sarcopenia screening questionnaires, such as NRS 2002 and SARC-F. The premise for study undertaken was the fact that many physicians are unaware of their patients’ nutritional status, and in even 40% of cancer patients malnutrition is overlooked by their physicians.  

Authors of manuscript confirm widely known fact that malnutrition and sarcopenia concern about 20-30% of in-patients and that they are related each other, to various comorbidities, and to their severity (e.g. Child stage). Authors do not test new questionnaires, and sometimes event torture data (Table 3, 4) showing relationships between malnutrition risk (NRS 2002 ≥ 3) and sarcopenia risk (SARC-F ≥ 4) and clinical scales of acute and chronic gastroenterological diseases severity (e.g. BISAP, Child) and comorbidities (e.g. coronary artery diseases, atrial fibrillation). It is obvious that the risks of malnutrition and sarcopenia corelate with severity of acute and chronic diseases.  Unfortunately, authors confuse the readers in Tables 1-5 that they compare patients with and without malnutrition and sarcopenia. Such diagnoses are impossible, because of they used screening tools only. Table 5 is presented in Turkish.

In results section authors should avoid to double data presentation in tables and in the text. Discussion should be shortened. Conclusion concerning need of treatment of patients with “malnutrition detected” is not supported by the results because authors diagnosed risk for malnutrition only. Therefore, manuscript title requires correction (e.g. Prevalence of Malnutrition Risk … ).

In summary, reviewed manuscript is not suitable for publication in present form. Because of screening for malnutrition and sarcopenia is obligatory for all in-patients in my country, the knowledge about prevalence of nutritional disorders in respective countries and clinical wards has no practical meaning. The only scientific value of this manuscript is to highlight the high prevalence of nutritional disorders risk (!) among in patients and that clinicians should be aware for nutritional disorders screening, prevent, and treatment.

Author Response

In their manuscript, authors raised very important issue which is prevalence risk of malnutrition and sarcopenia among in-patients. In this study authors choose heterogenous group of  1 047 patients hospitalized in the 36 gastroenterology clinics across 22 cities representing different geographical regions of Turkey. Authors used validated, widely used malnutrition and sarcopenia screening questionnaires, such as NRS 2002 and SARC-F. The premise for study undertaken was the fact that many physicians are unaware of their patients’ nutritional status, and in even 40% of cancer patients malnutrition is overlooked by their physicians. 

Answer 1: We sincerely thank the reviewer for their positive and thoughtful comments regarding our study. We are grateful that the reviewer recognized the clinical relevance of investigating the prevalence and risk factors of malnutrition and sarcopenia among gastroenterology inpatients. As noted, our study aimed to fill a significant gap in clinical practice, where nutritional assessment is often overlooked despite its major prognostic implications. We also appreciate the acknowledgment of our use of validated and widely accepted screening tools (NRS-2002 and SARC-F) and of the multicenter design including 36 gastroenterology clinics across different geographical regions of Turkey. We believe that these strengths contribute to the robustness and generalizability of our findings.

Authors of manuscript confirm widely known fact that malnutrition and sarcopenia concern about 20-30% of in-patients and that they are related each other, to various comorbidities, and to their severity (e.g. Child stage). Authors do not test new questionnaires, and sometimes event torture data (Table 3, 4) showing relationships between malnutrition risk (NRS 2002 ≥ 3) and sarcopenia risk (SARC-F ≥ 4) and clinical scales of acute and chronic gastroenterological diseases severity (e.g. BISAP, Child) and comorbidities (e.g. coronary artery diseases, atrial fibrillation). It is obvious that the risks of malnutrition and sarcopenia corelate with severity of acute and chronic diseases.  Unfortunately, authors confuse the readers in Tables 1-5 that they compare patients with and without malnutrition and sarcopenia. Such diagnoses are impossible, because of they used screening tools only. Table 5 is presented in Turkish.

Answer-2:We thank the reviewer for this important and constructive comment highlighting the distinction between diagnostic confirmation and risk screening. We fully agree that both NRS-2002 and SARC-F are screening tools that identify patients at risk of malnutrition and sarcopenia rather than providing a definitive diagnosis. In our study, we used these validated questionnaires in accordance with international ESPEN and EWGSOP2 recommendations, which emphasize the importance of early risk screening in hospitalized patients. To avoid any misunderstanding, we have carefully revised the manuscript to clarify throughout the text, tables, and figure legends that the terms “malnutrition” and “sarcopenia” refer to “risk of malnutrition (NRS-2002 ≥ 3)” and “risk of sarcopenia (SARC-F ≥ 4)”, respectively. All these revisions, including the updated Table 5, are highlighted in the revised manuscript for the reviewer’s convenience.

In results section authors should avoid to double data presentation in tables and in the text. Discussion should be shortened. Conclusion concerning need of treatment of patients with “malnutrition detected” is not supported by the results because authors diagnosed risk for malnutrition only. Therefore, manuscript title requires correction (e.g. Prevalence of Malnutrition Risk … ).

Answer-3: In accordance with these suggestions, we have revised the manuscript to avoid redundant data presentation by shortening the Results section and retaining detailed numerical values only in the tables. The Discussion section has also been condensed to improve focus and readability by removing repetitive explanations. Furthermore, we agree that our study assessed the risk of malnutrition and sarcopenia rather than confirmed diagnoses; therefore, the Conclusion was revised to emphasize the importance of systematic screening rather than treatment of diagnosed cases. The manuscript title has also been corrected to accurately reflect the study’s scope as “Prevalence and Clinical Associations of Malnutrition and Sarcopenia Risk in Gastroenterology Inpatients: A Multicenter Cross-Sectional Study in Turkey.” All these revisions have been highlighted in the revised version for the reviewer’s convenience.

In summary, reviewed manuscript is not suitable for publication in present form. Because of screening for malnutrition and sarcopenia is obligatory for all in-patients in my country, the knowledge about prevalence of nutritional disorders in respective countries and clinical wards has no practical meaning. The only scientific value of this manuscript is to highlight the high prevalence of nutritional disorders risk (!) among in patients and that clinicians should be aware for nutritional disorders screening, prevent, and treatment.

Answer-4: We respectfully acknowledge the reviewer’s opinion regarding the limited practical impact of prevalence data in settings where malnutrition and sarcopenia screening are already mandatory. However, we would like to emphasize that in Turkey, despite national recommendations, such screenings are not routinely implemented in daily clinical practice across gastroenterology wards. Therefore, our study provides the first national, multicenter evidence on the real-world rates of malnutrition and sarcopenia risk and their clinical associations among hospitalized gastroenterology patients. We believe that these findings have both scientific and practical value, as they identify high-risk patient groups, demonstrate variability across regions and disease types, and highlight the continuing gap between recommendations and practice.

Reviewer 2 Report

Comments and Suggestions for Authors

The manuscript addresses the prevalence and interrelationship of malnutrition and sarcopenia among hospitalized gastroenterology patients. The large multicenter cross-sectional study provides valuable insights. Several aspects should be improved as below:

Abstract
In the Methods section, please specify the statistical analyses performed or include key quantitative results.

Methods
Please provide information on whether clustering effects or adjustments for hospital-level variations across multiple centers were considered.

Discussion
Please include a clear limitations statement addressing the cross-sectional design’s inability to infer causality and the potential bias from self-reported SARC-F data.
The discussion is lengthy with excessive literature comparisons and may should be shortened to highlight the study’s novel findings.

Author Response

In the Methods section, please specify the statistical analyses performed or include key quantitative results.

Answer-1:This revised paragraph has been incorporated into the Methods section and is highlighted in the revised manuscript for the reviewer’s convenience.

Please provide information on whether clustering effects or adjustments for hospital-level variations across multiple centers were considered.

Answer-2:Since this was a cross-sectional, observational study in which each hospital contributed individual-level data collected during the same predefined week, all analyses were performed at the patient level. Given the descriptive and exploratory design of the study, no clustering or multilevel modeling was applied. However, to minimize center-related bias, we performed subgroup comparisons according to type of healthcare institution (university, training and research, or private hospital) and geographical region, as presented in Table 1. These variables were also evaluated in the regression analyses to account for institutional heterogeneity.

Please include a clear limitations statement addressing the cross-sectional design’s inability to infer causality and the potential bias from self-reported SARC-F data.
The discussion is lengthy with excessive literature comparisons and may should be shortened to highlight the study’s novel findings.

Answer-3: In accordance with the reviewer’s comment, we have revised and expanded the Limitations section to include a clear statement addressing the cross-sectional design’s inability to infer causality and the potential bias related to the self-reported SARC-F questionnaire. In addition, we have shortened and streamlined the Discussion section to focus on the study’s main findings and reduce excessive literature comparisons.

Reviewer 3 Report

Comments and Suggestions for Authors

The authors’ article focuses on the prevalence of sarcopenia/ malnutrition in various gastroenterological conditions.

The study do not include several conditions with an increased incidence and great impact on human health such obesity and MASLD associated sarcopenia, or celiac disease.

I consider enlarging the study group adding these categories of patients

  1. Sarcopenic obesity, described by the coexistence of high fat mass and low muscle mass or strength, due to chronic low-grade inflammation within the adipose tissue (especially visceral fat) , that releases inflammatory cytokines (e.g., TNF-α, IL-6), which promote muscle breakdown, associated with insulin resistance( also common in obesity), favored by physical inactivity, leading to muscle disuse and further loss of muscle mass, as well as myosteatosis (fat infiltration into muscle)

There is evidence that MASLD can worsen sarcopenia, by chronic inflammation, insulin resistance, and altered amino acid metabolism impair muscle growth. Furthermore, sarcopenia can worsen MASLD reducing muscle mass that leads to less glucose uptake and reduced insulin sensitivity, promoting hepatic fat accumulation.

Sarcopenia in MASLD patients is associated with faster progression to MASH, fibrosis, and liver-related complications.

Obesity and MASLD are not only associated with sarcopenia but may also mutually exacerbate it — forming part of a metabolic vicious cycle.

2.Celiac disease and other intestinal intolerances can cause secondary sarcopenia through a combination of malnutrition, inflammation, and metabolic disturbances, as  malabsorption and nutrient deficiency.

The intestinal mucosa damage reduces absorption of: proteins, vitamin D, calcium, iron, folate, and B12. These deficiencies impair muscle protein synthesis and neuromuscular function, leading to muscle wasting.

Celiac disease triggers immune-mediated inflammation (↑ TNF-α, IL-6, interferon-γ), cytokines that promote muscle catabolism and reduce muscle regeneration.

Chronic diarrhea, anorexia, and malabsorption can cause negative energy balance and muscle loss. Malnutrition leads to reduced IGF-1 and testosterone/estrogen, contributing to sarcopenia. Thus, in celiac disease (and other malabsorptive/intolerance conditions) we should consider screening for sarcopenia; current guidelines in CD recommend assessing nutritional status (malnutrition, sarcopenia, sarcopenic obesity) at diagnosis and follow-up.

Author Response

The study do not include several conditions with an increased incidence and great impact on human health such obesity and MASLD associated sarcopenia, or celiac disease.

I consider enlarging the study group adding these categories of patients

Sarcopenic obesity, described by the coexistence of high fat mass and low muscle mass or strength, due to chronic low-grade inflammation within the adipose tissue (especially visceral fat) , that releases inflammatory cytokines (e.g., TNF-α, IL-6), which promote muscle breakdown, associated with insulin resistance( also common in obesity), favored by physical inactivity, leading to muscle disuse and further loss of muscle mass, as well as myosteatosis (fat infiltration into muscle)

There is evidence that MASLD can worsen sarcopenia, by chronic inflammation, insulin resistance, and altered amino acid metabolism impair muscle growth. Furthermore, sarcopenia can worsen MASLD reducing muscle mass that leads to less glucose uptake and reduced insulin sensitivity, promoting hepatic fat accumulation.

Sarcopenia in MASLD patients is associated with faster progression to MASH, fibrosis, and liver-related complications.

Obesity and MASLD are not only associated with sarcopenia but may also mutually exacerbate it — forming part of a metabolic vicious cycle.

2.Celiac disease and other intestinal intolerances can cause secondary sarcopenia through a combination of malnutrition, inflammation, and metabolic disturbances, as  malabsorption and nutrient deficiency.

The intestinal mucosa damage reduces absorption of: proteins, vitamin D, calcium, iron, folate, and B12. These deficiencies impair muscle protein synthesis and neuromuscular function, leading to muscle wasting.

Celiac disease triggers immune-mediated inflammation (↑ TNF-α, IL-6, interferon-γ), cytokines that promote muscle catabolism and reduce muscle regeneration.

Chronic diarrhea, anorexia, and malabsorption can cause negative energy balance and muscle loss. Malnutrition leads to reduced IGF-1 and testosterone/estrogen, contributing to sarcopenia. Thus, in celiac disease (and other malabsorptive/intolerance conditions) we should consider screening for sarcopenia; current guidelines in CD recommend assessing nutritional status (malnutrition, sarcopenia, sarcopenic obesity) at diagnosis and follow-up.

Answer-1:We thank the reviewer for this detailed and thoughtful comment. We agree that conditions such as obesity, MASLD (Metabolic dysfunction-associated steatotic liver disease), and celiac disease are clinically relevant entities associated with nutritional disorders and sarcopenia. However, our study design included patients hospitalized during a predefined one-week period in gastroenterology clinics across Turkey. Therefore, some specific patient groups—such as those with obesity, MASLD-related sarcopenia, or celiac disease— have not been represented because these conditions often do not require inpatient hospitalization during that period. Our aim was to evaluate the real-world prevalence and risk of malnutrition and sarcopenia among currently hospitalized gastroenterology patients rather than outpatients or elective cases.

Round 2

Reviewer 1 Report

Comments and Suggestions for Authors

As I wrote in may first review, The only scientific value of this manuscript is to highlight the high prevalence of nutritional disorders risk among in patients and that clinicians should be aware for nutritional disorders screening, prevent, and treatment. 

Authors improved title, Tables, and explained more detaily the results of regression analysis, what improved my general rating of manuscript, and made it more suitable for publication. 

Author Response

As I wrote in may first review, The only scientific value of this manuscript is to highlight the high prevalence of nutritional disorders risk among in patients and that clinicians should be aware for nutritional disorders screening, prevent, and treatment.

Authors improved title, Tables, and explained more detaily the results of regression analysis, what improved my general rating of manuscript, and made it more suitable for publication.

Response: We thank the Reviewer for the positive reevaluation of our manuscript. Thank you for your constructive comments

Reviewer 3 Report

Comments and Suggestions for Authors

I suggest the authors to add the limitations of the study. Despite the limitations, I consider that the authors performed substantial work,  the study is useful and can be accepted for publication after minor revision.

This study has several limitations that should be considered when interpreting the findings. First, its cross-sectional design prevents the establishment of causal relationships between malnutrition, sarcopenia, and the associated clinical factors. Second, data were collected during a single week, which may not capture seasonal or temporal variations in patient characteristics or nutritional status. Third, the study population consisted solely of individuals attending gastroenterology clinics, which may limit the generalizability of the results to broader outpatient or community settings.

The heterogeneity of the underlying gastrointestinal and systemic diseases may also act as a confounding factor, despite adjustments made in regression analyses. Finally, potential inter-clinic variability across the 36 participating centers and unmeasured factors such as socioeconomic status, medication use, and physical activity may have influenced the observed associations.

Author Response

I suggest the authors to add the limitations of the study. Despite the limitations, I consider that the authors performed substantial work,  the study is useful and can be accepted for publication after minor revision.

This study has several limitations that should be considered when interpreting the findings. First, its cross-sectional design prevents the establishment of causal relationships between malnutrition, sarcopenia, and the associated clinical factors. Second, data were collected during a single week, which may not capture seasonal or temporal variations in patient characteristics or nutritional status. Third, the study population consisted solely of individuals attending gastroenterology clinics, which may limit the generalizability of the results to broader outpatient or community settings.

The heterogeneity of the underlying gastrointestinal and systemic diseases may also act as a confounding factor, despite adjustments made in regression analyses. Finally, potential inter-clinic variability across the 36 participating centers and unmeasured factors such as socioeconomic status, medication use, and physical activity may have influenced the observed associations.

Response: We thank the Reviewer for this valuable recommendation. In accordance with your suggestion, we have added a detailed “Limitations” paragraph to the end of the Discussion section. The newly added text has been highlighted in red in the revised manuscript.